# Evolution of Angiotensin Peptides and Peptidomimetics as Angiotensin II Receptor Type 2 (AT2) Receptor Agonists

**DOI:** 10.3390/biom10040649

**Published:** 2020-04-23

**Authors:** Silvana Vasile, Anders Hallberg, Jessica Sallander, Mathias Hallberg, Johan Åqvist, Hugo Gutiérrez-de-Terán

**Affiliations:** 1Sweden and Science for Life Laboratory, Department of Cell and Molecular Biology, BMC (H.G.T.), Biomedical Centre (BMC), Uppsala University, P.O. BOX 596, SE-751 24 Uppsala, Sweden; silvana.vasile@icm.uu.se (S.V.); jessica@sallander.com (J.S.); johan.aqvist@icm.uu.se (J.Å.); 2Department of Medicinal Chemistry, Division of Organic Pharmaceutical Chemistry, BMC, Uppsala University, P.O. Box 574, SE-751 23 Uppsala, Sweden; anders.hallberg@ilk.uu.se; 3The Beijer Laboratory, Department of Pharmaceutical Biosciences, Division of Biological Research on Drug Dependence, BMC, Uppsala University, P.O. Box 591, SE-751 24 Uppsala, Sweden; Mathias.Hallberg@farmbio.uu.se

**Keywords:** angiotensin, AT2R agonist, free energy perturbation (FEP), G-protein coupled receptor (GPCR), molecular dynamics (MD), peptidomimetics

## Abstract

Angiotensin II receptor type 1 and 2 (AT1R and AT2R) are two G-protein coupled receptors that mediate most biological functions of the octapeptide Angiotensin II (Ang II). AT2R is upregulated upon tissue damage and its activation by selective AT2R agonists has become a promising approach in the search for new classes of pharmaceutical agents. We herein analyzed the chemical evolution of AT2R agonists starting from octapeptides, through shorter peptides and peptidomimetics to the first drug-like AT2R-selective agonist, C21, which is in Phase II clinical trials and aimed for idiopathic pulmonary fibrosis. Based on the recent crystal structures of AT1R and AT2R in complex with sarile, we identified a common binding model for a series of 11 selected AT2R agonists, consisting of peptides and peptidomimetics of different length, affinity towards AT2R and selectivity versus AT1R. Subsequent molecular dynamics simulations and free energy perturbation (FEP) calculations of binding affinities allowed the identification of the bioactive conformation and common pharmacophoric points, responsible for the key interactions with the receptor, which are maintained by the drug-like agonists. The results of this study should be helpful and facilitate the search for improved and even more potent AT2R-selective drug-like agonists.

## 1. Introduction

Angiotensin II (Ang II, Asp^1^-Arg^2^-Val^3^-Tyr^4^-Ile^5^-His^6^-Pro^7^-Phe^8^) is an endogenous octapeptide that plays a central role in the Renin–Angiotensin signaling system. Ang II elicits a number of biological effects through activation of two specific G-protein coupled receptors (GPCR), namely the Angiotensin II receptor type 1 and 2 (AT1R and AT2R, respectively). Activation of the former results in pronounced hypertensive effects, a fact that promoted the development of new chemical entities interfering with this signaling pathway. Hence, drugs that block the formation of Ang II from angiotensinogen, e.g., captopril, inhibiting angiotensin converting enzyme (ACE) [1,2], or aliskiren that inhibits the aspartyl protease renin [3], were launched in 1978 and 2007, respectively. Losartan, the first selective AT1R antagonist marketed in 1995, is still widely used as an antihypertensive drug. It was the first of the family of sartans, AT1R antagonists containing a biaryl scaffold as common core structure [4,5]. On the contrary, activation of AT2R often results in opposite and beneficial pharmacological outcomes as compared to AT1R. These include pronounced vasodilatory, anti-fibrotic, anti-inflammatory, and neuroprotective effects [6]. Notably, the AT2R is strongly upregulated after tissue damage [7], such as at vascular [8] and neuronal injury [9], myocardial infarction [10], and brain ischemia [11]. Consequently, AT2R has in recent years emerged as an attractive target for future drug therapy [12,13,14,15]. Notably, AT1 and AT2 receptors only share 37% sequence identity.

The Hallberg lab discovered the first drug-like, selective AT2R agonist **C21** [16] (Figure 1), which, after successful characterization in a series of *in vitro* and *in vivo* models [17], is currently in Phase II clinical trials for idiopathic pulmonary fibrosis (IPF) [18]. The structure-activity relationships (SAR) around this scaffold was subsequently characterized, with the development of related selective and potent derivatives [16,19,20,21], and recent molecular models explained the observed selectivity for AT2R [22].

In effort to explore alternative scaffolds to those in the reported drug-like AT2R agonists, e.g., **C21**, we decided to refocus our attention on the small peptides that were proven to act as AT2R agonists [23]. In this context, it is essential to revisit the different biological outcomes elicited by Ang II and the bioactive peptides produced after its stepwise degradation *in vivo*. Peptide fragments often exhibit very different physiological properties as compared to their parent peptides [24,25]. Hence, the nociceptive substance P is degraded to a heptapeptide with anti-nociceptive effects [26,27] and degradation of Ang II via the heptapeptide angiotensin III (Ang III) (Arg^2^-Val^3^-Tyr^4^-Ile^5^-His^6^-Pro^7^-Phe^8^) results in the cognitive enhancer angiotensin IV (Ang IV) (Val^3^-Tyr^4^-Ile^5^-His^6^-Pro^7^-Phe^8^) that demonstrates a very low AT2R affinity [28,29]. The short-lived Ang IV inhibits insulin-regulated aminopeptidase (IRAP), resulting in cognitive enhancement in rat [28]. This observation motivated efforts to develop drug-like Ang IV peptidomimetics [30,31,32]. Contrary to Ang IV, Ang III (which only lacks the N-terminal Asp^1^ of Ang II) is an AT2R selective agonist and essentially equipotent to Ang II at this receptor. This observation indicates that the residue Asp^1^ in Ang II is not crucial for binding the AT2R [33,34]. Indeed it has been suggested that Ang III may play a central role as a predominant endogenous AT2R agonist, in particular in the kidney and possibly in the vasculature [35,36,37,38,39,40].

Encouraged by the recently disclosed crystal structure of AT2R bound to sarile (Sar^1^-Arg^2^-Val^3^-Tyr^4^-Ile^5^-His^6^-Pro^7^-Ile^8^, being Sar = sarcosine) **1** (Figure 1) [41], a proven agonist for this receptor AT2R [42], we conducted the first comprehensive analysis of the chemical evolution of peptides acting as AT2R agonists. This study was conceived to answer the following questions: What are the key pharmacophoric elements common to all AT2R agonists? What is the bioactive conformation of each peptide that satisfies the resulting pharmacophoric model? What are the essential key interactions that define an AT2R agonist? We present a retrospective analysis of the most salient peptides and peptidomimetics acting as AT2R agonists conducted through elucidation of the binding modes, via molecular docking, followed by molecular dynamics (MD) analysis of each pose to determine the key interactions. Subsequent binding affinity estimations with the free energy perturbation (FEP) method allowed explaining detailed structure-affinity relationships (SAR) of shorter pentapeptides and related mimetics, which further confirmed the feasibility of the binding model here presented. Finally, the gap between peptides and drug-like agonists is modelled herein on the basis of our calculations on compound **C21**, which should aid in the pursue of our long-term objective of enabling transformations of peptidic short-lived AT2R agonists into drug-like AT2R agonists with a long duration of action in vivo.

## 2. Materials and Methods

### 2.1. Protein Preparation

The crystal structures of AT1 and AT2 receptor in complex with sarile [41,43] were retrieved from the Protein DataBank (PDB codes 6DO1 and 5XJM). The structures were curated before computational simulations, including: (i) removing the co-crystalized proteins, i.e., nanobody Nb.AT110i1 in AT1R, and the thermostabilized apocytochrome b562 (BRIL) together with the fragment antibody Fab4A03, on the AT2R structure. (ii) Definition of the protonation state of titratable residues followed by optimization of the H-bond network and addition of hydrogen atoms. These steps were performed with the Maestro Protein Preparation Wizard (Schrödinger LCC, New York, NY, USA).

### 2.2. Docking and Molecular Dynamics (MD) Simulations

The three-dimensional (3D) structures of compounds **2**–**11** were initially generated and aligned to the experimental structure of sarile (**1**), as bound to the AT2R [41], using the Flexible Ligand Alignment tool within the same Maestro package (Schrödinger LCC, New York, NY, USA). The alignment was manually refined, when needed, to fulfill the hypothesis that common pharmacophoric elements should have similar interactions with the receptor as sarile. Similarly, the structure of compound **6** was aligned to sarile as co-crystallized to AT1R [43]. **C21** was manually docked to AT2R based on the docking models of C21-derivatives generated in our lab [22,44].

All receptor-ligand complexes were energy minimized with the Schrödinger’s Maestro MacroModel utility in Maestro (Schrödinger LCC, New York, NY, USA) using the Polak–Ribier conjugate gradient method with energetic convergence threshold of 0.05 kcal/mol and the OPLS3 force field. The complexes were then immersed in a pre-equilibrated phosphatidylcholine (POPC) membrane and equilibrated under periodic boundary conditions (PBC) using the PyMemDyn protocol [45]. Shortly, the structure is automatically embedded in a hexagonal prism-shaped box of pre-equilibrated membrane of POPC (1-palmitoyl-2-oleoyl phosphatidylcholine) lipids, with the TM bundle aligned to its vertical axis. This box is then soaked with bulk water and energy minimized using the OPLS-AA force field [46] for both the receptor and the peptide (missing parameters were obtained with Schrodinger’s ffld_server [47], combined with the Berger parameters for the lipids [48]). The same setup is used for a 2.5 nanosecond MD equilibration, where initial restraints on protein and ligand atoms are gradually released from 1000 to 200 kJ/molÅ^2^, followed by a 2.5 nanosecond run in which the 200 kJ/molÅ^2^ positional restraint is applied only in the C-alpha trace of the protein as described in detail in reference [45]. Thereafter, all the systems were subjected to 3 × 10 ns unrestrained MD simulations in GROMACS 5 [49], under the following conditions: isobaric NPT ensemble using a Nose-Hoover thermostat [50] with a target temperature of 310 K. Electrostatic interactions beyond a cutoff of 12 Å were estimated with the particle mesh Ewald (PME) method. Analyses of the MD runs were conducted with several GROMACS utilities and VMD [51].

### 2.3. Free Energy Perturbation (FEP) Calculations

The equilibrated complexes of AT2R with compounds **7**–**11** were then transferred to a spherical boundary system for free energy perturbation (FEP) calculations, performed with the software Q [52] using the force field parameters of OPLS 2015 [53]. A sphere with radius 25 Å, centered on the center of mass of each ligand, was defined. A 10 kcal/molÅ^2^ positional restraint was applied on solute atoms within the outer shell of the sphere (i.e., 23–25 Å from the center), while solvent atoms were subjected to polarization and radial restrains, using the surface constrained all-atom solvent (SCAAS) [52,54] model to mimic the properties of bulk water at the sphere surface. Solvent bond and angles were constrained using the SHAKE algorithm [55]. Atoms lying outside the simulation sphere were tightly constrained to their initial positions with a 200 kcal/molÅ^2^ force constant, and excluded from the calculation of non-bonded interactions. Long-range electrostatic interactions beyond a 10 Å cut-off were treated with the local reaction field (LRF) method [56], except for the atoms involved in the FEP transformations, for which no cut-off was applied. Ionizable residues near the boundary or outside the sphere were neutralized, to avoid artifacts due to missing dielectric screening [57]. The spherical systems were equilibrated with an initial heating phase lasting 31 ps, where temperature slowly raised up from 1 to 310 K, while the positional restraints initially applied to the solute atoms (10 kcal/molÅ^2^) were gradually released. It followed 0.1 ns unrestrained MD equilibration with the same parameters that would be later used for the FEP transformation: constant temperature of 310 K, bath coupling of 0.1 fs and time step of 1 fs. MD sampling consisted of 10 replica simulations, each starting with different random velocities, for each simulated state (i.e., solvated ligand and ligand-bound receptor), with the only difference between the two being a 0.5 kcal/molÅ^2^ force constantly applied to the geometrical center of the ligand for the simulations in water, in order to keep the molecule at the center of the sphere without hindering its rotation. Each independent simulation lasted for 1.53–4.08 ns (depending on the mutation), leading to a total sampling time of 2 (states) × 10 (replicates) x [1.53–4.08] (ns per replica per state) ≈ 55 ns. The standard error of the mean (SEM) was estimated in all cases from the 10 replicate simulations. Our FEP protocol for amino acid mutations has been described elsewhere [58,59,60]. Briefly, a given mutation of any residue to alanine was divided in several smaller subperturbations to allow for a smoother transition between the end-states. Three steps were introduced for groups of atoms (charge groups) starting with the group with the highest topological distance (number of atoms) from the protein backbone: (1) removal of partial charges per charge group, (2) introduction of a soft core van der Waals potential, and (3) full annihilation of the involved atom(s). The last step included the introduction of the Cβ hydrogen atom of the alanine residue. For non-alanine mutations, initial models of mutant receptors were created with PyMol, using the most probable rotamer of the mutated residue, and a double thermodynamic cycle was joined (i.e., WT → Ala and mut → Ala), meaning that in these cases double simulation time was needed. To compare calculated with the experimental binding affinity shifts between two ligands, A and B, we converted the experimental *Ki* values into ΔΔG using:
(1)$$\Delta \Delta G=RTln\frac{K_{i}^{A}}{K_{i}^{B}}$$

## 3. Results

### 3.1. Selection of the AT2R Agonists Dataset and Initial Docking

In this work, we focused on the elucidation of the SAR and binding mode of eleven AT2R agonists (peptides of different lengths and peptidomimetics). Compounds **1**–**11** represent the only proven agonists for AT2R, where the agonism has been determined with the neurite outgrowth assay in all cases [17]. The starting point of our analysis is the octapeptide sarile (**1**, Figure 1), a potent and non-selective AT2R agonist that was used decades ago as an antihypertensive “angiotensin antagonist” in the clinic, and that was recently co-crystallized with both AT2R and AT1R [41,43]. This compound has been characterized as a partial agonist on AT1R (see [61] and references therein), and it is reasonable to speculate the same pharmacological profile on AT2R. However, the pharmacological data on AT2R based on neurite outgrowth assay cannot differentiate between partial or full agonism [42]. Sarile comprises a N-terminal sarcosine residue for improvement of metabolic stability in vivo [62], and an Ile^8^ rather than the Phe^8^ present in Ang II, as the C-terminal residue. The equipotency of Ang II and sarile (**1**), combined with the high affinity shown by Ang III towards AT2R, demonstrates that Asp^1^ is not crucial for binding [33,34]. CGP42112A (**2**), a selective AT2R agonist extensively used to characterize this receptor [63,64,65], contains the same His-Pro-Ile C-terminal sequence as **1**, while it bears a unique branched N-terminal structure as opposed to the rest of the linear peptides studied here (Figure 1) [66]. Compounds **3**–**6** (Figure 1) represent a series of conformationally restrained analogues where the central Tyr^4^-Ile^5^ dipeptide fragment of Ang II is replaced by different γ-turn mimicking scaffolds. The agonists comprising a benzodiazepine γ-turn mimic exhibit excellent AT2 affinity in the low nanomolar range [67], while compound **6**, containing a tri-substituted phenyl ring, exhibits a somewhat reduced affinity [68]. Moreover, it has been shown that replacement of the Arg^2^ sidechain of **3** by a methyl group, i.e., the corresponding Ala^2^ derivative, has a deleterious effect on affinity demonstrating that the guanidine group of **3** is of outmost importance for AT2R binding [69]. Additionally, a Glu scan of Ang II revealed that a Glu^2^ derivative furnishes a 100-fold lower affinity to AT2R [34]. Finally, we selected a series of truncated AT2R agonists **7**–**11** of five amino acid residues or less and with either a phenylalanine or a isoleucine amino acid residue in the C-terminal, showing interesting variations in affinity (Figure 1) [70,71]. It is well established that removal of three amino acid residues from the n-terminal of Ang II analogues could result in favored selectivity for AT2R [72]. The impact of the C-terminal sidechain is clearly illustrated in a comparison of the phenylalanine derivatives **7** and **9** with their isoleucine analogues **8** and **10**, respectively, the latter exhibiting more than 10-fold higher affinity to AT2R and high selectivity versus AT1R. It was somewhat unexpected that the affinity could be improved by replacement of the N-acetyl-Tyr-Ile dipeptide residue in the N-terminal of **8** by a 4-hydroxy diphenylmethane moiety (**10**). Interestingly, the presence of a basic sidechain on position 2 (Arg^2^) seems crucial for AT2R binding affinity of octapeptides, while the shorter pentapeptides and analogues (**7**–**11**) are all devoid of a corresponding guanidino group, and still exhibit good to fair AT2R affinity. Finally, Figure 1 shows the structure of the AT2R selective drug-like **C21** [17], to enable structural comparisons with the shortest peptides.

The new experimental structure of the AT2R in complex with sarile (**1**) [41] represents an excellent opportunity to finally understand the SAR and the evolution of peptidic and peptidomimetic AT2R agonists in the context of ligand-receptor interactions. To achieve this goal, we initially docked compounds **2**–**11** using the backbone of the co-crystallized sarile as a template. The sidechains of equivalent residues were modelled in the same rotameric state as in sarile, retaining the corresponding interactions with the receptor. While in most cases the analogy between peptide fragments is trivial, some cases deserve a closer look and generation of working hypotheses that need to be tested in the models. This was particularly true for compound **2** (CGP42112A), which presents a guanidine group on a branched-like structure mimicking the sidechain of Arg^2^ in Ang II. As discussed before, the available SAR of linear octapeptides suggests that Arg^2^ plays a key role for their affinity to AT2R. In the sarile-AT2R crystal structure this sidechain forms salt bridges with the aspartic acid residues D279^6.58^ and D297^7.32^, interactions, which could be easily mimicked in the docking complexes generated for the octapeptide analogues **3**–**6**, encompassing different turn mimicking scaffolds. Consequently, we generated a pose of compound **2** where its guanidine group is positioned in a similar region, maintaining analogous interactions with the same two aspartic acid residues (see Figure 2).

### 3.2. Molecular Dynamics Simulations of Octapeptides and Mimetics (Compounds **1**–**6**)

The experimental structure of AT2R with the octapeptide sarile (**1**), as well as the modeled complexes with analogues **2**–**6**, were subject to 3 × 10 ns MD simulations in an atomistic model of the membrane, solvated under periodic boundary conditions. Figure 3 shows a generic representation of the peptide-receptor complex, indicating the occurrence and frequency along the MD sampling of the common receptor-peptide interactions. A representative structure of each peptide-receptor complex is depicted in Appendix A, and the coordinates provided as Appendix A.

Starting from the most buried C-terminal fragment, the simulations confirm the stability of the salt bridge between the carboxy terminus and K215^5.42^ for all peptides. The C-terminus is at the same time involved in intramolecular hydrogen bonds with the sidechain of Tyr^4^, which is lining between ECL2 (M197, F199) and TM5 (S208^5.35^, I211^5.38^), with the exception of **2**, where the corresponding Tyr^2^ sidechain faces TM6 and forms tight internal hydrogen bonds with the arginine mimic. Additionally, the orientation of the sidechain at the C-terminus is different depending on the nature of the amino acid residue at this position: the aliphatic Ile^8^ present in compounds **1** and **2** is accommodated in a hydrophobic pocket within helices TM3 (L124^3.32^, T125^3.33^, M128^3.36^), TM6 (F272^6.51^), and TM7 (Y304^7.39^ and F308^7.43^). While the C-terminal Phe of **6** is located in the same hydrophobic pocket, the remaining octapeptide analogues (**3**–**5**) place the aromatic Phe sidechain in position 8 buried between TM3 (T125^3.33^, M128^3.36^, F129^3.37^), TM4 (T178^4.40^), and TM5 (K215^5.42^) (Figure 3B). The sidechain of Phe8 occupies a similar space as the second ring of the biphenyl substituent of two antagonists co-crystallized in the inactive structures of AT2R (Appendix A). Despite the fact that we are comparing inactive and active conformations of the receptor, this observation is in line with the typical design of antagonists assuming that they share some pharmacophoric points with agonists. The effect of the corresponding Ile/Phe substitution is further examined below in the series of shorter peptides (**7**–**11**) with the aid of free energy perturbation techniques. Following the peptide sequence, the proline ring (Pro^7^) is consistently interacting with W100^2.60^ within the series, while the His^6^-Pro^7^ amide bond is hydrogen bonding with the guanidinium group of R182^4.64^ (Figure 3C). As discussed above, the potential H-bond of His^6^ with D297^7.32^ in the AT2R is at sub-optimal distance in the crystal structure with **1**, and our MD simulations show indeed variable occupancies of this interaction, frequently mediated by a water molecule. In compound **2**, this histidine is hindered by the guanidinium moiety, resulting in an alternative intramolecular interaction with the C-terminus (Appendix A), while mainly interacting with D297^7.32^ (**3**, **6**) or Y104^2.65^ (**6**), see Figure 3. The γ-turn mimicking fragments in **3**–**6** are lining towards the ECL2 surface (detected by frequent vdW contacts with residues I187, I196, M197 and Y204^5.31^, Figure 3B). From these, compounds **3**, **4** and **6** retain the analogous sidechain of Val^3^ in **1**, but only in **4** it was found to occupy the same binding subpocket. This could indeed explain the relative lower affinity of **5**, an analogue of **4** lacking the Val sidechain. Within this group, the less potent compound is **6** bearing a more rigid benzene scaffold as a γ-turn mimetic. This substitution does not allow hydrogen bonding between M197 and the backbone of Tyr^4^, otherwise frequent in **3**–**5**, where the benzodiazepine carbonyl mimics the backbone of a tyrosine residue. The extracellular area of the binding site is dominated by the salt bridges frequently observed between Arg^2^ in **1**, **2**–**6** with the pair of aspartic residues in TM6 (D279^6.58^) and TM7 (D297^7.32^), supported by electron-π interactions with W283^6.62^. Finally, the N-terminus was modelled in the open extracellular area and is mostly solvent-exposed. Particularly interesting is compound **2**, which presents an N-terminal pyridine ring located between ECL2 and TM6, (W283^6.62^). In agreement with the experimental SAR, the MD simulations showed that the zwitterionic N-terminus aspartic acid fragment (compounds **3**–**6**) is not involved in particular interactions with the receptor and instead is solvent accessible.

### 3.3. Selectivity between AT1R and AT2R

The crystal structure of sarile in complex with AT1R [43] allows a direct comparison of the binding mode of peptides with different selectivity profiles versus AT1 and AT2 receptors. The sequence homology between the two receptors is as low as 37%, with some differences located in the agonist binding site [22]. In addition, the bulkiness or charge of amino acid sidechain may play an important role for AT2R selectivity of the ligands. Nevertheless, the binding mode of sarile (**1**) in AT1R and AT2R is very similar (Figure 4A and Appendix A), with the main differences located at the extracellular side. As opposed to ATR2, the N-terminus of AT1 is folded towards ECL2, creating a binding crevice for the sarcosine residue of **1** and a deeper location of Arg2 between TM6 and TM7, as compared to AT2R. Consequently, the MD simulations show extensive salt bridges between Arg^2^ and D263^6.58^ and D281^7.32^, stronger than the corresponding interactions in AT2R (Figure 3). However, sarile is equipotent regarding affinity to the AT1 and AT2 receptors (*K_i_* 0.16 nM and 0.18 nM, respectively [68]), and the differences observed might be consequence of crystal packing, due to the use of different experimental conditions.

More notable is compound **6**, showing a 3-fold affinity reduction for AT1R as compared to AT2R (*K*_i_ = 30.3 nM and 9.8 nM for AT1R and AT2R, respectively [68]). Analysis of the corresponding MD simulations suggests that this difference could arise from the dissimilar orientation of the C-terminal Phe residue of this compound, which in the AT1R would be surrounded by L112^3.36^ (Met in AT2), K199^5.42^, H256^6.51^ (Phe in AT2), and Y292^7.43^ (Phe in AT2). This pocket, which in both AT1R and AT2R hosts the Ile of **1**, appears more closed in the AT2R receptor, where the corresponding C-terminus Phe sidechain in the ligand folds back in a deeper hydrophobic pocket between TM3, TM4, and TM5, which could explain the gain in affinity for this receptor (Figure 4B).

### 3.4. FEP Simulations of the Pentapeptides and Mimetics **7**–**11**

Enhanced AT2R selectivity upon N-terminal truncation of Ang II analogues was already observed almost three decades ago [72]. The effect of replacing the aromatic Phe sidechain at the C-terminus by an aliphatic sidechain, which generally improves AT2R affinities [66], is amplified in the shortened peptides. This trend is clearly observed in the truncated pentapeptides **7**–**8** and related peptidomimetics **9**–**11** (Figure 1) [70,71]. Here, the replacement by Ile of the C-terminal Phe leads to a 10 to 20 fold gain in affinity (i.e., compare **8** and **10** with their analogues, **7** and **9**, respectively). These two pairs also demonstrate the beneficial effect of replacing the N-terminal dipeptide residue (Tyr-Ile), with its two amide bonds, by a benzene ring scaffold to which a 4-hydroxy-benzyl group is attached to replace the Tyr sidechain of **7** and **8** (c.f. the benzene ring scaffold in **6**). Further, the potency of the resulting non selective AT2R agonist **9** was further improved by introducing the amino group at the 4-hydroxy-diphenylmethane scaffold, which in addition improves the affinity towards the AT1R [23].

We herein quantified the effect of these chemical modifications on the pentapeptides and analogues by simulating the corresponding sidechain modifications via our recently developed free energy perturbation (FEP) protocols [73]. The results (Table 1) show overall good convergence (average SEM of 0.56 kcal/mol) and accuracy, quantified as a mean absolute error (MAE) of 0.85 kcal/mol, in line with previous applications of this technique [73].

The transformations between **8** → **7** and **10** → **9** both involve an Ile → Phe mutation at the C-terminus. Since the binding site of each sidechain is predicted to be different (see above for the long peptides **1**–**6**), the strategy followed was to independently model each compound in the binding site and annihilate the C-terminal sidechain to the common Ala intermediate. With this strategy one can estimate the overall effect by joining the two resulting thermodynamic cycles, as described in Methods and references [58,59,60,74]. The corresponding FEP calculations estimate an improvement in the binding affinity for the two Ile-containing ligands, providing qualitative support for the docking hypothesis indicating that Phe and Ile sidechains sit in different subpockets in the AT2R (Figure 5). The interactions of a C-terminal Ile sidechain, located within the hydrophobic pocket defined by helices TM3, TM6, and TM7, are most energetically favored for binding than the accommodation of a Phe sidechain within TM3, TM4, and TM5 (Figure 5). Finally, we also studied the effect of introducing an additional amino group on the benzene ring used as a γ-turn mimic in compound **9**, yielding compound **11**. This compound could be seen as a truncated version of compound **6**, indeed with a similar binding affinity for AT2R (Table 1). However, in this case we found a binding pose where the free amino group can interact with D297^7.32^, and the N-terminal tyrosyl is instead pointing towards the extracellular region of the receptor, providing a rationale for the slight increase in binding affinity of the aminated (**11**) version (see Table 1).

### 3.5. Putative Binding Mode of C21

A putative binding mode of **C21** was generated by manual docking, guided by the structural information of the sarile-AT2R complex, and assisted by our previous modelling that explained the SAR of this structural class of compounds [22]. Subsequently MD refinement revealed the key receptor ligand interactions (Figure 6), dominated by electrostatic interactions of the sulfonyl carbamate with the positively charged sidechains of K215^5.42^ and R182^4.64^, complemented by the hydrogen bond with T125^3.33^. A second electrostatic anchoring point is located on the imidazole ring, consistently forming a hydrogen bond with Y104^2.65^. Finally, the isobutyl group in this agonist scaffold is enclosed in a hydrophobic pocket formed by TM2, TM3, TM6, and TM7 (see Figure 6A), which is receptor-conformation specific thus connected to the agonist/antagonist the activity of the **C21** derivatives. Indeed, this model was recently used in the course of our AT2R project to explain the SAR of a novel series of **C21** derivatives as AT2R antagonists [44].

A comparison of the binding mode of **C21** with the peptides and peptide mimetics studied in this paper is quite informative to understand and unify our knowledge of the SAR of the different scaffolds. **C21** binds deeper in the receptor as compared to sarile (**1**, Figure 6B), with the isobutyl group displaying additional hydrophobic interactions in the binding crevice that undergoes conformational change during receptor activation [43,75,76]. Notably, the imidazole heterocycle of **C21** is still found in the same area as the imidazole ring of His^6^ of sarile, in both cases showing hydrogen bonds with Y104^2.65^. Finally, the sulfonyl carbamate functional group, designed as the carboxylic acid mimic for the C-terminus of the peptides, presents analogous electrostatic interactions with K215^5.42^ and R182^4.64^, with the butyl substituent extending parallel to the sidechain of the former (Figure 6B).

## 4. Discussion

We revised the evolution of peptides and peptidomimetics towards small molecule agonists of the AT2 receptor, taking advantage of the new structural information of this receptor. The focus was put on disclosing structure-affinity relationships and key protein-ligand interactions, by means of molecular dynamics and free energy calculations. The large amount of data accumulated along decades of chemical development was here reduced to a carefully selected dataset of octapeptides containing various turn mimetics (**1**-**6**), as well as shorter pentapeptides truncated at the N-terminal and chemically modified (**7**-**11**), all proven to act as AT2R agonists. Both series include γ-turn mimetics, the role of which in direct ligand-receptor interactions remained elusive. The effect of peptide truncation and the incorporation of turn-mimicking groups can now be illuminated with the recent crystal structures of sarile (**1**) with both the AT1R [43] and AT2R [41]. Using this valuable experimental information, we built a consistent model for the binding of the common fragments and analogous regions of all the peptides and mimetics within this series. In the longer octapeptides, the first key interactions are found in the extracellular area of the receptor, by means of ionic interactions between Arg^2^ in the peptide and aspartic acids in TM6 (D279^6.58^) and TM7 (D297^7.32^). We also showed how the hydrophobic central part of the peptides, replaced by the benzodiazepine or phenyl ring moieties in the γ-turn mimic peptidomimetics, makes extensive contacts with residues of the ECL2. Here, we revealed the important modulatory role of the residue Val^3^, which conveniently oriented seems to be important for the affinity of the octapeptides, see sarile (**1**) and **4**.

The consistency of the binding mode of octapeptides was assessed by MD simulations of each AT2R-agonist complex, creating conserved fingerprints of protein-ligand interactions (Figure 3 and Figure 4). It followed the docking, by analogy of the C-terminal fragment, of a series of pentapeptides and analogues (**7**–**11**), where the role of different chemical modifications could be examined in more detail. Indeed, the effect of structural variability on binding affinity is amplified on shorter peptides which, together with the advantage of working for the first time with a crystal structure of the AT2R-agonist bound receptor conformation, allowed us to quantify the shifts in binding free energy along the series of shorter peptides **7**–**11**. We initially focused on the interactions of the C-terminal residue, where a change of the natural Phe^8^ in Ang II for Ile is consistently associated with an increase in binding affinity. While the binding site of Ile^8^ was revealed in the corresponding X-ray structure of sarile with AT2R, the positioning of a Phe is not necessarily the same. Instead, our binding model shows that the Phe sidechain would occupy the same area as the biphenyl ring of AT2R antagonists, in a pocket defined by TM3, TM4, and TM5 [75,76]. The only exception is the lower affinity peptidomimetic **6**, where the C-terminal Phe occupies the same binding pocket as Ile^8^ of sarile (**1**). It should be noted that, during the preparation of this manuscript, two new crystal structures of AT1R [61] and AT2R [77] in complex with the endogenous peptide Ang II were released. Appendix A compares our starting binding mode for compounds **3-6**, containing a Phe in the C-terminus, with the experimental binding mode of Ang II in complex with AT2R. It can be appreciated that the rotameric state proposed in our models is the same as the Phe^8^ in Ang II, and essentially this sidechain occupies the same binding crevice, despite of the change in the orientation of the AT2R sidechain Met128^3.36^ observed in the new Ang II-bound structure [77].

Analysis of our MD simulations based on the sarile-bound AT2R crystal structure indicates that the sidechain of K215^5.42^ would be acting as a “gatekeeper”, preventing spontaneous rotameric transition between the two alternative binding regions of the C-terminal sidechain. As previously indicated, the new AT2R-Ang II structure reveals a change in the sidechain orientation of Met128^3.36^ associated to the nature of the C-terminal chain in the co-crystalized ligand [77]. Whether this change is a response to the different pharmacological profile of Ang II and sarile (a proven partial agonist on AT1R) is a hypothesis that deserves further exploration, using for instance the thermodynamic cycles strategy recently proposed by our lab [78].

Nevertheless, the role of Phe/Ile mutation on the binding affinities of AT2R agonists was further tested with FEP simulations. Energetic comparison of **8** → **7** and **10** → **9** shows that the affinity gains due to the Phe to Ile mutation is specifically captured by a model that considers the different binding orientation of the two sidechains, while the same simulations considering a common conformation results in the loss of correlation with experiments (Appendix A). Within these series, compounds **9**–**11** are shortened versions of the peptidomimetic **6**, all containing a phenyl moiety as a γ-turn mimic, or in the case of compound **11** an aniline, with the amino group mimicking to some degree the amide bond extending to the amino terminal in compound **6**. The equipotency of these two compounds was intriguing, in particular since the loss of the free amino group (i.e., compound **9**) affects the affinity. Our simulations suggest that the role of this amino group is heavily influenced by the existence of larger substitutions on the amino terminal side. Thus, in compound **11** this amino group makes a specific interaction with D297^7.32^, a finding could guide the design of novel derivatives that better explore this interaction.

The release of the crystal structure of sarile in complex with AT1R allowed a preliminary analysis of the selectivity between AT1R and AT2R, using two non-selective peptides of our dataset, compounds **1** and **6**. Contrary to sarile (**1**), which is equipotent at the two receptors, compound **6** shows a 3-fold reduction of affinity for AT1R as compared to AT2R, which again can be connected to the orientation of the C-terminal sidechain. For this compound, the C-terminal Phe shows a different orientation within the two receptors: while in AT2R this aromatic residue occupies a pocket formed by TM3, TM6 and TM7 (as all the peptides with a C-terminal Ile), in AT1R this pocket is more closed and the Phe is surrounded by TM3, TM4, and TM5. To further test the viability of these observations, we redocked compound **6** on the new Ang II-bound ATR crystal structures. As shown in Appendix A, the compound achieves a similar binding mode despite the change in the rotameric state of the sidechain in position 3.36.

Finally, we compared the binding mode of peptides and peptidomimetics with the previously proposed binding mode of the low-molecular-weight agonist **C21**. The binding mode of this compound is anchored by electrostatic interactions of the central sulphonyl carbamate with charged sidechains of TM2 and TM5, analogous to its role as C-terminus mimic of the peptidic agonists. Interestingly, the butyloxy group is extending the binding crevice of with the sidechain if Phe^8^ in the peptides, making hydrophobic interactions with the aliphatic part of the sidechain of K215^5.42^. This would be compatible with the optimal size for activity of the butyl as alkylic substituent in this position, since shorter chains and longer or bulkier results in lower affinity [16,19,20,21]. The isobutyl group would be located in a deeper hydrophobic cavity within the TM region, which is specifically opened for the active receptor conformation explaining why this group is so characteristic of this series of AT2R agonists. The model, thus, supports the structure-activity relationships for these compounds, revealing that the relative position of the imidazole and the isobutyl chain as a landmark of AT2 agonism [17,79]. However, to explain in detail the observed impact on affinity and functionality after small stereochemical alterations of AT2R ligands encompassing rigidified isobutyl groups remains a challenge [80].

## 5. Conclusions

Our analysis of the evolution of octapeptides and peptidomimetics towards shorter pentapeptides, in the light of the last crystal structure of the agonist-bound AT2 receptor, shows for the first time a structural landscape of peptide-receptor interactions. The analysis not only explains the SAR observed along decades of peptide to small molecule evolution, but also depicts potential points for chemical intervention in the optimization of these molecules. Thus, we revealed the effect of some classical strategies on peptide optimization on peptide-AT2R interactions, such as the length of the peptide or the introduction of γ-turn mimics. Then, we moved on to the shorter peptidomimetics and quantified the role on ligand binding affinities of some chemical modifications the C-terminal residue or the γ-turn mimic moiety. Finally, a comparison with the binding orientation of the small-molecule agonist scaffold **C21**, on the light of the recent crystal structures, allowed identification of the essential protein-ligand interactions, which are minimal to achieve agonist activity, which should facilitate the structure-based design of novel low molecular weight agonists for this receptor.

## Figures and Tables

**Figure 1 biomolecules-10-00649-f001:**
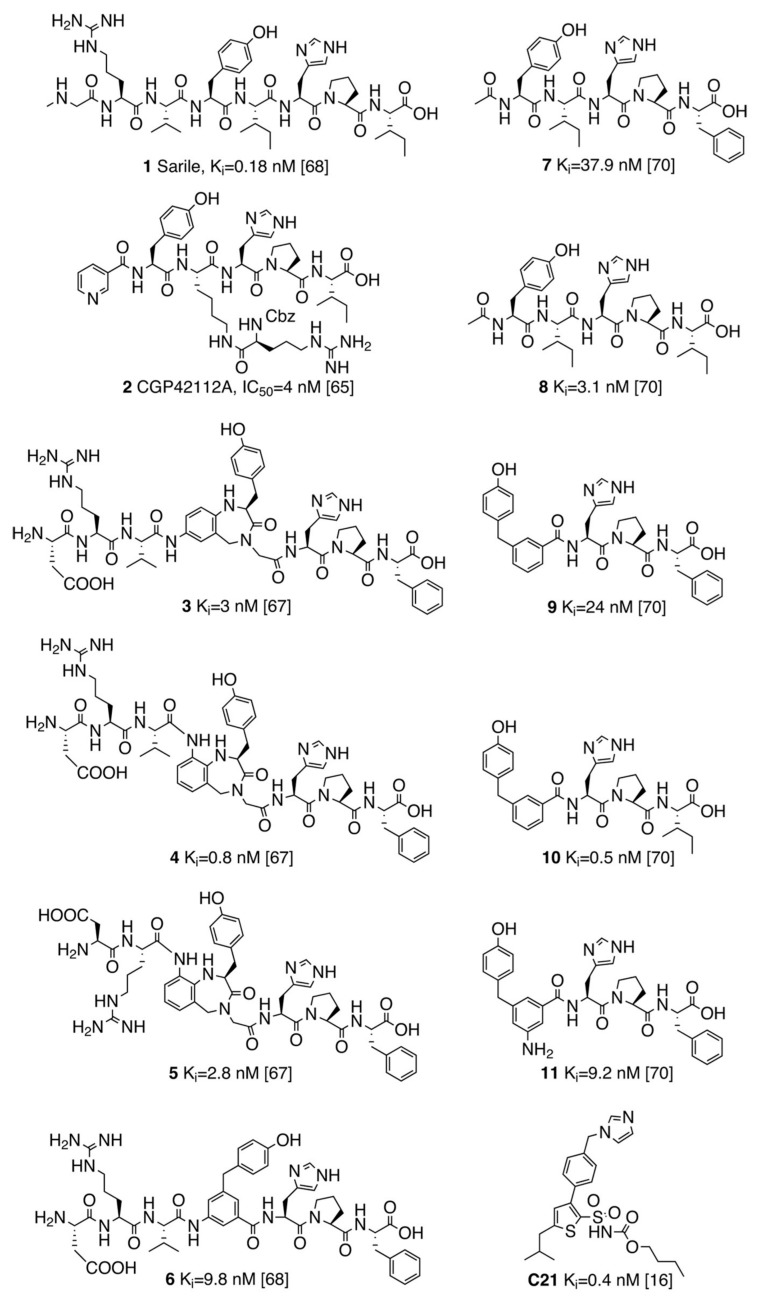
Structures of all the peptides considered in this study. Left column collects the octapeptides and analogues. On the right column, truncated pentapeptides and analogues are depicted. Binding affinities to angiotensin II receptor type 2 (AT2R) are shown (K_i_ and IC_50_ values, in nM) with references. The corresponding binding affinities to angiotensin II receptor type 1 (AT1R) are: **1** (K_i_ 0.16 nM), **2** (IC_50_ > 10,000 nM), **3** (K_i_ > 10,000 nM), **4** (K_i_ > 10,000 nM), **5** (K_i_ > 10,000 nM), **6** (K_i_ 30,3 nM), **7** (K_i_ > 10,000 nM), **8** (K_i_ > 10,000 nM), **9** (K_i_ 196 nM), **10** (K_i_ > 10,000 nM), **11** (K_i_ > 10,000 nM).

**Figure 2 biomolecules-10-00649-f002:**
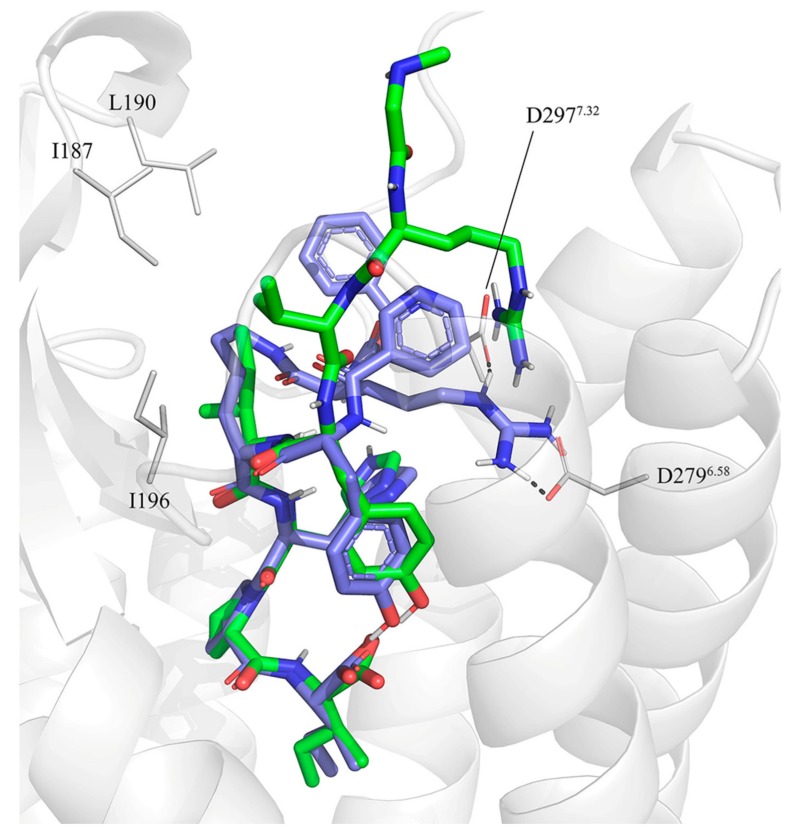
Initial docking of CGP42112A, compound **2** (violet) superimposed to the crystal structure of sarile (**1**) (green) in complex with AT2R. Dashed lines indicate the conserved salt bridge between Arg^2^ and D279^6.58^ and D297^7.32^ (lines). The lipophilic sidechains of AT2R surrounding the equivalents of Ile^5^ and Val^3^ in **1** are also shown.

**Figure 3 biomolecules-10-00649-f003:**
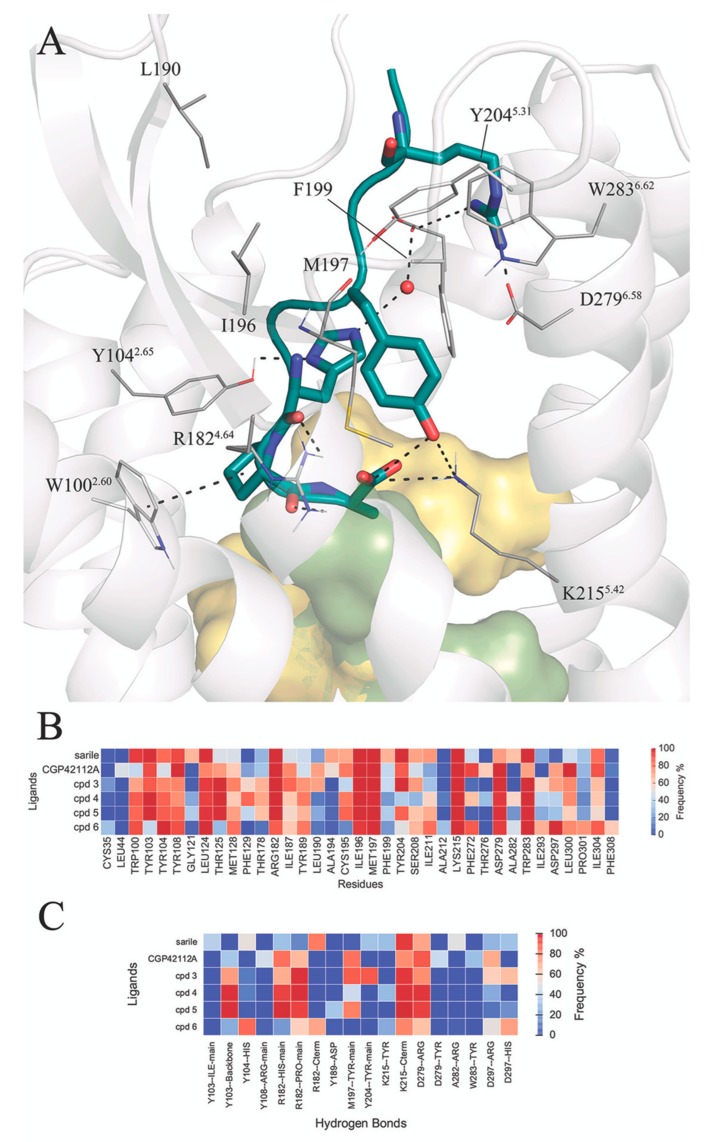
(**A**) Schematic representation of the octapeptide analogues **1**–**6** in the AT2R binding site. The hydrophobic pocket surrounding the C-terminus is depicted as a surface (yellow for Ile^8^, green for Phe^8^), the receptor residues in contact with the common sidechains of the peptides (see panel B) are explicitly represented and hydrogen bonds with high frequency (see panel C). The heat-maps represent atom contacts (**B**) and hydrogen bonds (**C**) occurring more than 30% of the simulation time between at least one peptide and the AT2R.

**Figure 4 biomolecules-10-00649-f004:**
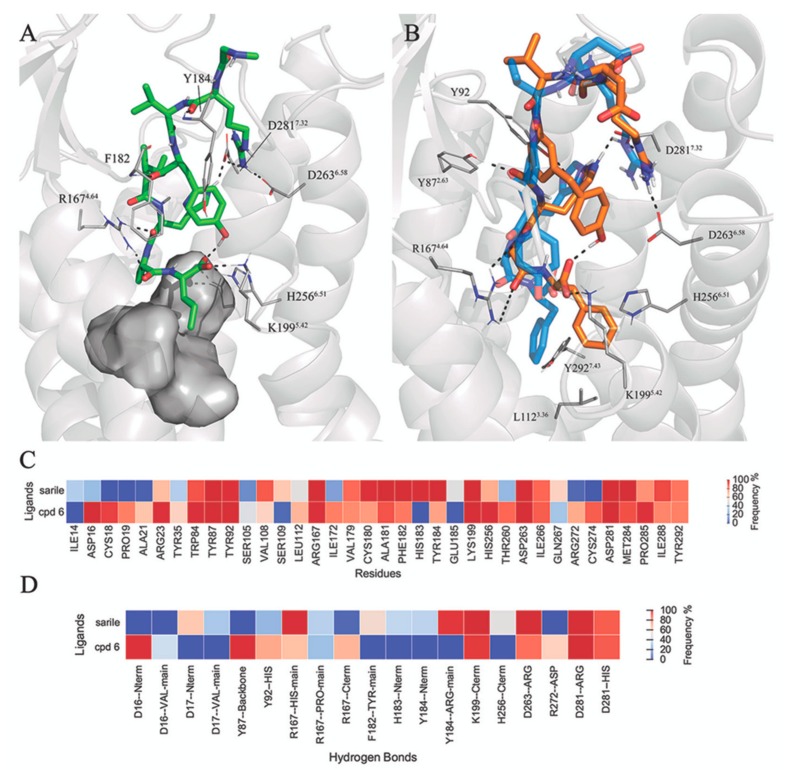
(**A**) Binding pocket of sarile (**1**) in AT1R. (**B**) Binding mode of **6** (orange sticks) in AT1R (gray ribbons), with the binding mode in AT2R overlaid in the background in blue. The heat-maps represent atom contacts (**C**) and hydrogen bonds (**D**) occurring more than 30% of the simulation time between **1**, **6** and AT1R.

**Figure 5 biomolecules-10-00649-f005:**
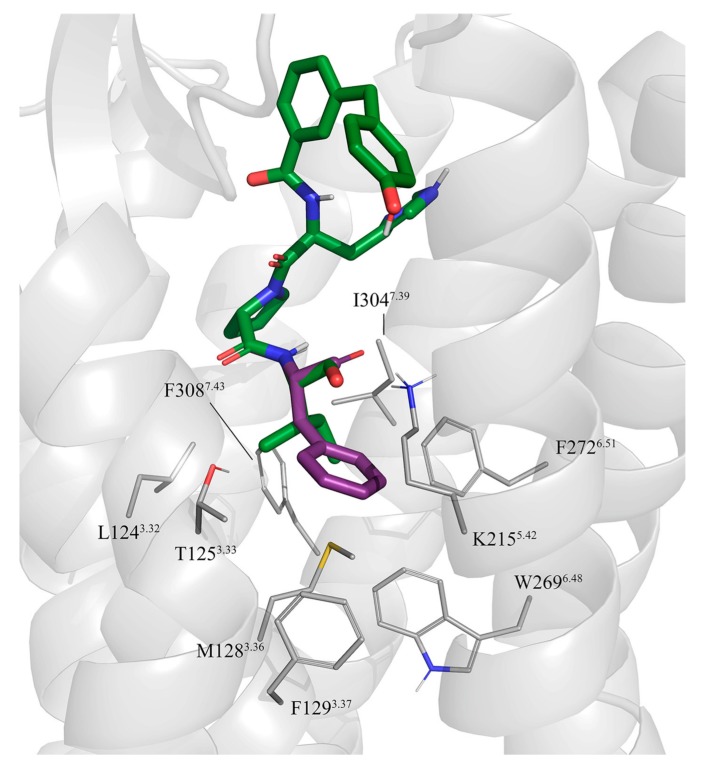
Comparison of the binding mode of **9** (purple sticks) and **10** (dark green sticks), with depiction of the sidechains surrounding the C-termini. The different orientations of the Ile/Phe could explain the change in affinity observed for these two compounds.

**Figure 6 biomolecules-10-00649-f006:**
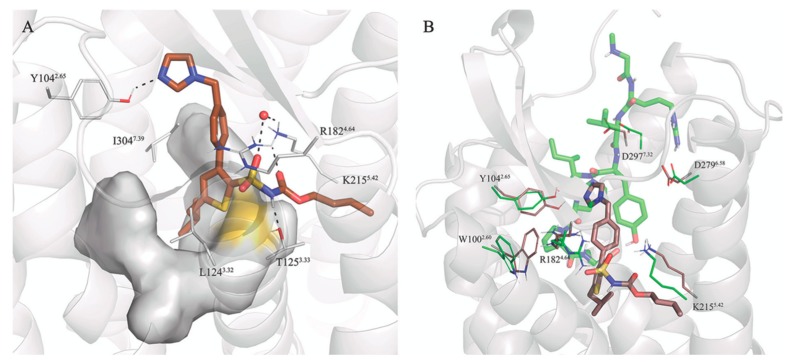
(**A**) Binding mode of **C21** (brown sticks) to the AT2R. The grey surface represents the hydrophobic pocket surrounding the isobutyl group of **C21**. (**B**) Comparison of the binding modes of **C21** (brown sticks) and sarile (**1**, green semi-transparent sticks). The main AT2R residues showing interactions with each ligand are shown in lines, and hydrogen bonds indicated with dotted lines.

**Table 1 biomolecules-10-00649-t001:** Relative binding affinity (in terms of experimental and calculated shifts in the free energy of binding) between three pairs of AT2 agonists.

Ligand Pair	Chemical Modification	ΔΔG_exp_ ± SEM (kcal/mol)	ΔΔG_calc_ ± SEM (kcal/mol)
**8** → **7**	Ile → Phe	1.54 ± 0.06	0.97 ± 0.64
**10** → **9**	Ile → Phe	2.38 ± 0.07	0.55 ± 0.67
**11** → **9**	Aniline → Phenyl	0.59 ± 0.06	0.46 ± 0.36

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
