# Peer review of "Evolution of Angiotensin Peptides and Peptidomimetics as Angiotensin II Receptor Type 2 (AT2) Receptor Agonists"

_biomolecules, 2020, doi:10.3390/biom10040649_

Round 1

Reviewer 1 Report

The manuscript by Vasile et al. is presented as a review on Evolution of Angiotensin Peptides and 2 Peptidomimetics as AT2 Receptor Agonists, with a title and an introduction which do not fit  to the study described.

Briefly, it is a very standard computational study, docking and molecular dynamics, using the crystal structures of AT1 and AT2 in complex with sarile and eleven already published compounds. Undoubtfully the study suggests the binding mode of these compounds with the target protein and gives some information on the putative interaction of the compounds with AT2. However, it should be taken into account that the crystal structures of AT1 and AT2 receptor in complex with sarile already represent a solid starting point for drug design.

The binding hypothesis proposed in this manuscript is not supported by experimental data (site-directed mutagenesis or the synthesis of a small number of targeted compounds) thus in my opinion the manuscript should be rejected.

In addition, an extensive editing of the English language and style is suggested.

Author Response

Reviewer: Briefly, it is a very standard computational study, docking and molecular dynamics, using the crystal structures of AT1 and AT2 in complex with sarile and eleven already published compounds. Undoubtfully the study suggests the binding mode of these compounds with the target protein and gives some information on the putative interaction of the compounds with AT2. However, it should be taken into account that the crystal structures of AT1 and AT2 receptor in complex with sarile already represent a solid starting point for drug design.

Response: The referee seems to miss that, in addition to “standard” molecular docking and molecular dynamics, our study includes free energy perturbation (FEP) calculations a technique that not many groups in the world perform on GPCRs. In the FEP section (lines 338-380, Table 1 and Table S1, Figure 5) we reinforce our modeling proposal with correctly reproduced relative binding affinities for the shorter peptides. Moreover, these simulations anticipate a different binding mode for the C-terminal Phe as compared to Leu, such as in Sarile. Importantly, this hypothesis is now validated with the recent crystal structures of ATR-AngII (see response to next point). Another key point of novelty of the present study is that the carefully selected compounds in Figure 1, represent the only proven agonists for AT2R, where the agonism has been determined with the neurite outgrowth assay in all cases. This sentence is now included in the manuscript (lines 180-182).

Reviewer: The binding hypothesis proposed in this manuscript is not supported by experimental data (site-directed mutagenesis or the synthesis of a small number of targeted compounds) thus in my opinion the manuscript should be rejected.

Response: The new structures of AngII-bound ATRs, released during the submission of this manuscript (as kindly indicated by referee 3) now provide experimental validation to our binding mode hypothesis for the C-terminal residue, as discussed above. We have updated this information with new references 61 and 77, and a new paragraph in the discussion section (lines 445-451).

Reviewer 2 Report

The author applied molecular docking, MD simulation and FEP calculations to identify the selective binding mechanism of a series of agonists to AT2R.  The current results are helpful and facilitate the search for designing potent AT2R-selective agonists. The methods used are accurate and the results are solid. Only one concern is that 10ns MD simulations seems too short for me. It is better to extend the MD simulation a bit. Otherwise, the author should check the convergence of the simulations. It is better to provide the RMSD of each system in Supporting information.

Author Response

Referee: Only one concern is that 10ns MD simulations seems too short for me. It is better to extend the MD simulation a bit. Otherwise, the author should check the convergence of the simulations. It is better to provide the RMSD of each system in Supporting information.

Response: The RMSD of each system, expressed as an average over the 3x10 ns replicates in each case, is now included as Figure S13 in the supporting information.

Reviewer 3 Report

In this manuscript entitled "Evolution of Angiotensin Peptides and Peptidomimetics as AT2 Receptor Agonists", the authors carried out the MD simulation and FEP calculation of C21 and 11 selective AT2R agonist based on sarile bound AT2R and AT1R structure. From these results, important interaction of AT2R selective agonist and the receptor were demonstrated. AT2R selective antagonist is very important drug for the related diseases. In this point, this report is considered very helpful. However, several potential areas need to improved:

Recent studies have shown that the endogenous peptide AngII binds to AT1R and AT2R (Structure. 2019 doi: 10.1016 / j.str.2019.12.003, Science. 2020 doi: 10.0.1126 / science.aay9813). These studies show the structural evidence that AngII is a full agonist, whereas sarile is a partial agonist of ATR. Also, structural comparisons of sarile- or AngII-bound AT2R / AT1R show that the bottom of the ligand binding cavity is different clearly. These structural differences can affect biological activity. In particular, the side chain orientation in Met1283.36 (L112 in AT1R) has been shown to play an important role for the receptor activation. The side chain orientation of 3.36 position seems the same for sarile and other antagonists bound ATR. However, its orientation is clearly different in the Ang II bound receptors. Therefore, it should need to be justified that the use of the partial agonist sarile, but not Ang II, as a starting point for your study (but I agree with the results of the authors). In this report, authors showed that the difference in the C-terminus of the peptide agonists causes the difference in the side chain orientation of the hydrophobic amino acids that consist the bottom of the ligand binding cavity. This may affect the intensity for receptor activation, just like relationship of AngII and sarile. In this point, it is neceaary that author's model is compared with AngII bound ATRs.

The authors have to describe not only the affinity for AT2R but also the affinity for AT1R and specify the source of the Ki values (Figure 1, p5 line 193-194, p10 line 291-292).

The authors claim that the C-terminus orientation of Phe8 in octapeptide 3-5 is same as two AngII inhibitor bound AT2R structures (line245-249). However, in these AT2R structure, the activation motifs (DRY, PIF and NPxxY) are same in sarile bound AT2R structure. The authors consider these octapeptides as antagonists? This description is confusing.

The “Selectivity between AT1R and AT2R” section is most important for development of AT2R specific agonist. Therefore, it is important to compare the structure between author's and AngII bound ATRs structures, not only sarile bound ATRs. In particular, it is better that the docking simulation for the ligand 6 is performed using the AngII bound AT1R and AT2R (Figure 4). Because the C-terminal amino acid is Phe. The authors note that the sequence homology between AT1R and AT2R is low. The bulkiness or charge of amino acid side-chain may play an important role for AT2R selectivity of the ligands.

Author Response

Referee: Recent studies have shown that the endogenous peptide AngII binds to AT1R and AT2R (Structure. 2019 doi: 10.1016 / j.str.2019.12.003, Science. 2020 doi: 10.0.1126 / science.aay9813).

Response: We thank the referee for noticing these very recent crystal structures of the AT receptors with the natural agonist AngII. These structures were indeed released around the during the submission of this manuscript and therefore not considered in the initial submission. The revised version of the manuscript includes these new references (61 and 77) together with some additional analysis on the light of this new information, see below.

These studies show the structural evidence that AngII is a full agonist, whereas sarile is a partial agonist of ATR.

We agree on the partial agonism of sarile on AT1R, from the data in Wingler et al. Science. 2020 doi: 10.0.1126 / science.aay9813 (and references therein), and as there suggested it is also reasonable to anticipate the same behavior on AT2R. However, a note of caution should be added since the pharmacological readout on AT2R is at the present moment based on a neurite outgrowth assay, which cannot differentiate between partial or full agonism (see ref 42 in our manuscript). We have included this comment on lines 185-188 and will discuss on the structural hypothesis of partial versus full agonism below.

Also, structural comparisons of sarile- or AngII-bound AT2R / AT1R show that the bottom of the ligand binding cavity is different clearly. These structural differences can affect biological activity. In particular, the side chain orientation in Met1283.36 (L112 in AT1R) has been shown to play an important role for the receptor activation. The side chain orientation of 3.36 position seems the same for sarile and other antagonists bound ATR. However, its orientation is clearly different in the Ang II bound receptors. Therefore, it should need to be justified that the use of the partial agonist sarile, but not Ang II, as a starting point for your study (but I agree with the results of the authors).

This is indeed an interesting question. We have added a new paragraph on the discussion section, lines 438-442, relative to future studies planed to reveal the role of Thr3.36 in the recognition on molecules with different pharmacological profile. In response to a point below, we also compared compound 6 in the two structures obtaining similar binding modes.

In this report, authors showed that the difference in the C-terminus of the peptide agonists causes the difference in the side chain orientation of the hydrophobic amino acids that consist the bottom of the ligand binding cavity. This may affect the intensity for receptor activation, just like relationship of AngII and sarile. In this point, it is neceaary that author's model is compared with AngII bound ATRs.

It is worth noting that our binding orientation of compounds 3-6 (containing a C-terminal Phe) in the sarile-bound structure adopts the same rotamer as the corresponding Phe in AngII-ATRs structures, even though our models were generated without considering that information. We now provide this comparison in a supplementary figure (Figure S11), and added a comment in this sense (lines 428-434).

The authors have to describe not only the affinity for AT2R but also the affinity for AT1R and specify the source of the Ki values (Figure 1, p5 line 193-194, p10 line 291-292).

The sources of the Ki values were already cited throughout the manuscript, but thanks to this comment of the referee we noted that they were incomplete, besides not explicitly added on the Figure. The present version includes the AT1R data on the legend on Figure 1, and all references in Figure 1 including the missing reference for compound 6 (new reference 68). Note that all are Ki values except for compound 2 (CGP42112A) where only IC50 is available.

The authors claim that the C-terminus orientation of Phe8 in octapeptide 3-5 is same as two AngII inhibitor bound AT2R structures (line245-249). However, in these AT2R structure, the activation motifs (DRY, PIF and NPxxY) are same in sarile bound AT2R structure. The authors consider these octapeptides as antagonists? This description is confusing.

The comparison with biphenyl antagonists has also been clarified:, and the new paragraph in lines 264-268 reads “The sidechain of Phe8 occupies a similar space as the second ring of the biphenyl substituent of two antagonists co-crystallized in the inactive structures of AT2R (Figure S8). Despite the fact that we are comparing inactive and active conformations of the receptor, this observation is in line with the typical design of antagonists assuming that they share some pharmacophoric points with agonists.”

The “Selectivity between AT1R and AT2R” section is most important for development of AT2R specific agonist. Therefore, it is important to compare the structure between author's and AngII bound ATRs structures, not only sarile bound ATRs. In particular, it is better that the docking simulation for the ligand 6 is performed using the AngII bound AT1R and AT2R (Figure 4). Because the C-terminal amino acid is Phe.

The C-terminal Phe of compound 6 in our docking, using sarile-bound structures, is similar to that of Phe8 in the recently released AngII structures. Additional docking on the new structures, performed following the suggestion of the referee, provide essentially the same binding mode for this fragment (see lines 463-466 and new Figure S12 in the Supplementary Information).

The authors note that the sequence homology between AT1R and AT2R is low. The bulkiness or charge of amino acid side-chain may play an important role for AT2R selectivity of the ligands.

We appreciate this comment, which we have included in the discussion of our manuscript (lines 297-298)

Round 2

Reviewer 1 Report

In my opinion the manuscript should  be rejected because it is still far from being suitable for the journal.

Reviewer 3 Report

The authors addressed most of my queries satisfactory and the manuscript has been greatly improved. So, I think that this manuscript OK to be accepted.